# Perceptions of institutional performance and compliance to non-pharmaceutical interventions: How performance perceptions and policy compliance affect public health in a decentralized health system

**Marco Antonio Catussi Paschoalotto**[ID][1]*, **Eduardo Polena Pacheco Araújo Costa**[1], **Sara Valente de Almeida**[2], **Joana Cima**[3], **Joana Gomes da Costa**[4], **João Vasco Santos**[ID][5,6,7], **Claudia Souza Passador**[8], **João Luiz Passador**[8], **Pedro Pita Barros**[1]

1 Nova School of Business and Economics, Nova University of Lisbon, Lisbon, Portugal, 2 School of Public Health, Imperial College London, London, England, 3 Centre for Research in Economics and Management (NIPE), University of Minho, Braga, Portugal, 4 Center for Economics and Finance, School of Economics and Management, University of Porto, Porto, Portugal, 5 MEDCIDS—Department of Community Medicine, Information and Health Decision Sciences, Faculty of Medicine, University of Porto, Porto, Portugal, 6 CINTESIS—Center for Health Technology and Services Research, Faculty of Medicine, University of Porto, Porto, Portugal, 7 Public Health Unit, ACES Grande Porto VIII—Espinho/Gaia, ARS Norte, Porto, Portugal, 8 School of Economics, Business Administration and Accounting at Ribeirão Preto, University of São Paulo, Ribeirão Preto, Brazil

* marcocatussi@gmail.com

**Data Availability Statement:** The dataset is anonymous and does not allow the identification of

## Abstract

Trust in institutions is a key driver to shape population attitudes and behavior, such as compliance of non-pharmaceutical interventions (NPI). During the COVID-19 pandemic, this was fundamental and its compliance was supported by governmental and non-governmental institutions. Nevertheless, the situation of political polarization in some countries with decentralized health systems increased the difficulty of such interventions. This study analyzes the association between non-pharmaceutical interventions' compliance and individual perception regarding institutions' performance during the COVID-19 pandemic in Brazil. A web survey was conducted in Brazil between November 2020 and February 2021. Bivariate analysis and ordered logit regressions were performed to assess the association between NPIs compliance and perceived institutions' performance. Results suggest a negative association between NPIs' compliance and Federal Government and Ministry of health perceived performance, which may reflect the political positioning of the respondents. Moreover, we find a positive association between NPI compliance and the perceived performance of the remaining institutions (state government, federal supreme court, national congress, WHO, media and SUS). Our contribution goes beyond the study of a relationship between non-pharmaceutical interventions' compliance and institutions' performance, by pointing out the importance of subnational and local governmental spheres in a decentralized health system, as well as highlighting the importance of social communication based on health organizations' information and scientific institutions.

individual respondents. However, based on the informed consent signed by respondents and on the authorization granted by NOVA School of Business and Economics Ethics Committee, data cannot be shared in a public repository to respect data protection regulation. A minimal data set for replication of the study findings is available upon request to the NOVA School of Business and Economics Ethics Committee (research. office@novasbe.pt) and the corresponding author. All relevant data are within the manuscript and its Supporting Information files.

**Funding:** This report is independent research funded by Fundação para a Ciência e a Tecnologia (UIDB/00124/2020, UIDP/00124/2020 and Social Sciences DataLab - PINFRA/22209/2016), POR Lisboa and POR Norte (Social Sciences DataLab, PINFRA/22209/2016). The funders had no role in study design, data collection and analysis, decision to publish, or preparation of the manuscript.

**Competing interests:** The authors have declared that no competing interests exist.

# Introduction

Previous studies regarding a population's attitudes and behavior have shown an association between trust in institutions and perceived performance [1–3]. Additionally, it is well established that individual political positioning can influence the societal decision-making process, and ultimately the compliance with public health interventions [4–6]. In the last three years, we have seen this phenomenon during the COVID-19 pandemic. Compliance with non-pharmaceutical interventions (NPIs) seems to be related with the perception of performance of some institutions as well as with individuals' political positioning [3, 5, 7]. As i) NPIs we consider individual (e.g. face mask), environmental (e.g. ventilation of spaces) and population (e.g. social distancing and stay-at-home/quarantine measures) interventions, and as ii) compliance we consider the self-reported level of compliance (Never, Rarely, Frequently and Always) with NPIs [3].

Although the vaccination process is widely diffused, the COVID-19 pandemic is not over, and non-pharmaceutical interventions are still important [8–11]. Despite the importance of adopting NPIs to manage the evolution of the pandemic, some countries have shown difficulties in their implementation and population compliance. Hence, it is essential to understand the factors associated with NPIs' compliance [12, 13].

Political factors and social media have an impact on NPIs compliance levels [2, 14–17]. The amount and frequency with which COVID-19 related news were and are spread can affect perceptions on institutions' performance and influence the population's compliance with NPIs [7, 18, 19]. This influence can vary through different levels of governmental institutions, ranging from federal government [9, 20–22], to the executive, legislative and judiciary powers of the Brazilian State [17, 23, 24], as well as international institutions, e.g. World Health Organization [25–28].

Regarding the American continent for example, it was described that the USA, Brazil, and Mexico had different performances in restricting their populations' mobility through different public health approaches, but all experienced problems of coordination [29]. These problems were mainly seen in Brazil and Mexico [30]. Mexico did not present an uniform national response, with the states varying in their implementation of NPIs, compromising the national health system governance [31]. Brazil also did not present a national coordinated response in the public health measures to combat the COVID-19 pandemic, with the virus spreading differently across the states [32, 33].

Until August 2022, more than 600 million COVID-19 cases and 6.4 million deaths had been recorded worldwide. Brazil ranks in 4th place in number of cases (over 34 million) and 2nd in total deaths (over 680 thousand) [34]. Brazil is considered an upper middle-income country, with great inequalities among states and municipalities. Its federal structure implies shared responsibilities of public policies between states and municipalities [27, 32, 35, 36]. The Brazilian Unified Health System (SUS–*Sistema Único de Saúde*) is the national institution responsible for public health services in Brazil, including vaccination and epidemiological programs [25, 37].

Throughout most of the COVID-19 pandemic (and until new presidential elections), Brazil witnessed an attitude of denial from the nation's former President himself, who discouraged NPIs application [38, 39]. According to the literature, one of the former President Jair Bolsonaro's arguments to retain his followers during these times was to provide misinformation through social media platforms [6, 40, 41]. In response to this attitude, the judiciary and legislative powers, the Federal Supreme Court (STF) and the National Congress, took political measures to supplement the federal government action [42–44]. The states and municipalities acquired more autonomy to act against the COVID-19 pandemic, due to a Federal Supreme

Court (STF) decision, and the number of social programs to provide financial support to the Brazilian society increased, due to National Congress emergency legislation [27, 32, 42, 45]. The states' governors and respective state health secretaries aligned their responses, supported by the National Council of State Health Secretaries (CONASS), to implement public health measures to combat the COVID-19 spread [33].

For the reasons presented above, Brazil can be considered as an interesting case study to understand the association between institutions performance and behavioral compliance with government measures, such as the NPIs in the COVID-19 pandemic, in a context of polarized political society and decentralized health system [4, 33, 46].

We investigate the importance of institutions' performance, while studying the association between NPI compliance and individual perception of the institutions' performance itself. For this purpose, we rely on an online convenient national survey to explore how political positioning and the perception of institutions' performance influence the individual's willingness to comply with NPIs.

This paper stresses the need of a discussion on public health and the decision-making process, while highlighting the role of the i) subnational and local governments in a decentralized health system, ii) open scientific evidence, such as support from WHO, and iii) SUS to use social communication in public health emergency situations.

## Materials and methods

The study relied on an online survey developed by the research team and disseminated among the Brazilian population. The survey was approved on 23 November 2020, by the Research Ethics Committee at NOVA School of Business and Economics (Portugal) through a Scientific Council's president letter. In the online survey's first page, the participants read and accepted the research characteristics and information, assurances of anonymity assurance, data protection, and a consent form.

### Data collection

The invitation to participate in the survey was shared on social networks, such as Facebook®, WhatsApp®, LinkedIn®, Instagram® and mailing lists. The target populations included individuals present in all Brazilian states, from different social sectors and backgrounds. Data were collected between 25 November 2020, and 11 January 2021, through an online web survey, created and distributed using Qualtrics software. The time interval regarding data collection was important because: i) it was applied before the second largest (and deadliest) COVID-19 wave, ii) the survey was responded immediately before the introduction of the first COVID-19 vaccine in Brazil, CoronaVac, in March 2021, iii) it marked a change in the strategy addressing COVID-19, shifting from NPIs to vaccination, and iv) it marked the consolidation of the population' perception towards public health measures, which is seen after 9 months of introduction.

Table 1 provides a comparison between our sample and national averages. The sample is characterized by 1,621 valid responses, with respondents coming from most of the Brazilian States (88.9%) and States' Capitals (74.1%). We also see an over-representation of individuals living in São Paulo State (67%). Although this is thus not a representative sample, it is considered a convenient and informative sample with some sociodemographic conditions being aligned with the Brazilian population. Besides the sociodemographic characterization, we have introduced the political position characteristics of the sample.

In addition to the sociodemographic conditions (location, gender, age and education), data on institutions' performance perception, NPIs' compliance level and individual political

**Table 1. Sample characteristics.**

| Variable | Survey Data | | National Data |
|---|---|---|---|
| **States and Municipalities (number)** * | | | |
| States | 24 | | 27 |
| States' Capitals | 20 | | 27 |
| Municipalities | 263 | | 5,570 |
| **Gender (%)** | | | |
| Male | 37.9 | | 48.2 |
| Female | 61.7 | | 51.8 |
| Other/Not responding preference | 0.4 | | |
| **Age (%)** | | | |
| < 18 years | 0.9 | < 14 years | 24.1 |
| 19 to 25 years | 30.6 | 15 to 64 years | 68.5 |
| 26 to 32 years | 20.9 | | |
| 33 to 45 years | 27.4 | | |
| 46 to 64 years | 18.4 | | |
| 65 to 79 years | 1.9 | 65 years > | 7.4 |
| 80 years > | 0.1 | | |
| **Education (%)** | | | |
| Elementary School | 0.5 | | 55.8 |
| High School | 14.4 | | 30.1 |
| University–Degree | 40.5 | | 14.1 |
| University–MBAs and Specializations | 20.8 | | |
| University–Master | 14.1 | | |
| University—Doctorate | 9.7 | | |
| **Political Position (%)** | | | |
| 1 –Far Left | 2.3 | | NA |
| 2 | 14.3 | | |
| 3 | 19.7 | | |
| 4—Center | 21.6 | | |
| 5 | 11.7 | | |
| 6 | 7.7 | | |
| 7 –Far Right | 2.7 | | |
| No answer | 20 | | |

* Brazil has more than 75% of its municipalities characterized as "small" (< 25.000 inhabitants), which generates difficulties in achieving a representative sample [47].

positioning were collected, with no influence among the categories (S1 Table). Respondents were asked to evaluate (on a qualitative scale ranging from "very bad", "bad", "average", "good", to "very good") the performance of the following institutions in the COVID-19 pandemic: Federal Government, State Government, Municipal Government, Unified Health System (SUS), Ministry of Health, Brazilian Federal Supreme Court (STF), National Congress, World Health Organization and the Media institutions. We have considered Media as an institution due to its relevance and influence on society regarding the pandemic and NPIs measures [48].

Regarding NPIs compliance, respondents were asked to disclose their self-reported level of compliance (never, rarely, frequently and always) with the following NPIs: Face Mask Use, Social Distancing and Stay Home (when possible). To identify the respondents' political

preferences, they were asked to report their perceived position on a scale from 1 (far left) to 7 (far right) (S1 Table).

## Data analysis

This study sought to reveal the association between compliance with NPI related measures (Face Mask Use, Social Distancing, and Stay Home) and the population's perception regarding institutions' performance. We used bivariate analysis and estimated ordered logit models to analyze the associations, always using ordered dependent variables (1-Never; 2-Rarely; 3-Often; 4-Usually). These models are based on latent variables Y*, as follows [49]:

$$Y^* = \beta X + \varepsilon \tag{1}$$

where X is a matrix (N x K) containing a set of variables as explained above (i.e., gender, age, education, perception of institutions' performance, and political position), and $\varepsilon$ corresponds to the error term with a logistic distribution. The unknown cut-points ($\alpha$) are defined as follows, where both parameters $\alpha$ and $\beta$ are estimated by maximum likelihood.

$$Y = 1 \; if \; Y^* \leq \alpha_1$$

$$Y = 2 \; if \; \alpha_1 < Y^* \leq \alpha_2$$

$$\ldots$$

$$Y = 4 \; if \; Y^* > \alpha_3 \tag{2}$$

The analysis was conducted using the statistical software Stata 16.

## Results

### Government spheres (federal/state/municipal)

Fig 1 displays the association between compliance with NPIs and the perceived performance for a set of governmental institutions at the federal, state and municipal levels.

Regarding the use of face masks (panel A), individuals typically report very high compliance levels, regardless of their perception of different governmental institutions. However, it is still possible to observe a gradient between these two variables. In fact, the proportion of individuals using face masks seems to decrease as the perception regarding the federal government performance improves. Opposite results are found for the state level. Moreover, no major gradient is observed at the municipal level.

Social distancing (panel B) and stay at home recommendations (panel C) follow a similar pattern to face mask use. As individuals have a better perception regarding the Federal Government performance, their level of compliance tends to decrease. Results present an inverse pattern at the state level. Again, no major gradient is identified at the municipality levels.

### Powers' tripartition (ministry/congress/ federal supreme court)

Fig 2 displays the association between compliance with NPIs and the perceived performance for a set of national institutions representing the different branches of power.

The use of face masks (panel A) seems to be negatively related with the perception of the Ministry of Health performance. This follows the same result as the one found for the federal government. Contrarily, compliance with the use of face masks increases with the positive perception of the performance of both the National Congress and Federal Supreme Court. Similar

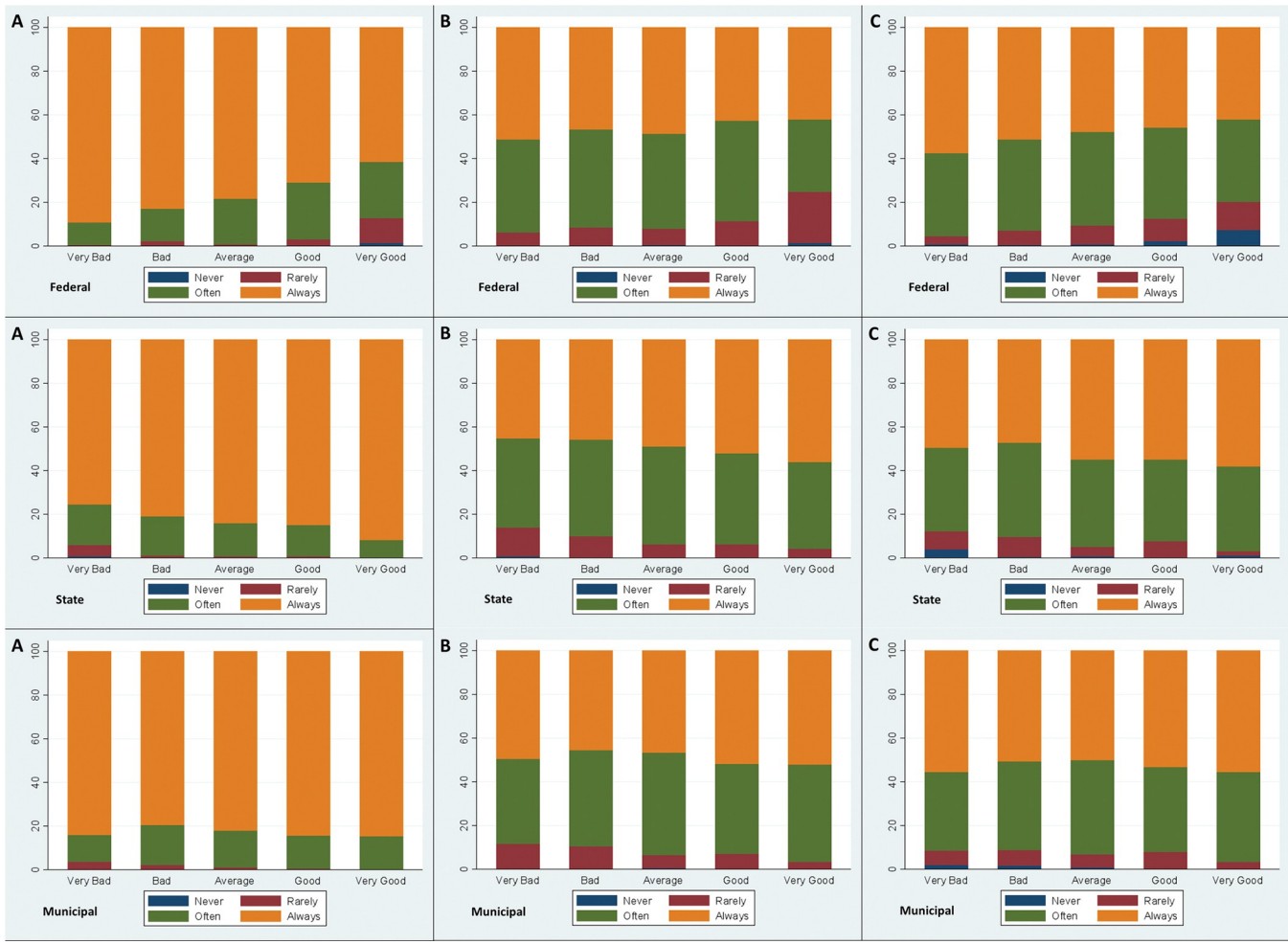

**Fig 1.** Federal, State and Municipal government spheres x Face Mask (A), Social Distancing (B) and Stay Home (C).

results are observed when looking at both the social distance (panel B) and stay at home (panel C) recommendations. However, one should note that the proportion of individuals reporting to always comply with these recommendations is significantly lower than the ones reporting to always use face masks.

## Organizations (WHO/SUS/Media)

Fig 3 displays the association between compliance with NPIs and the perceived performance for a set of non-governmental institutions.

Perceptions and compliance when considering the WHO, SUS and Media performance present a similar behavioral pattern. Overall, compliance with NPIs increases as the perception regarding the performance of these institutions increases. This pattern is common to all NPIs.

## Regressions

We considered an ordered logit model to study the potential determinants of compliance towards (A) face mask use, (B) social distancing, and (C) stay at home. We also present the institutions' performance odds and tables of conditional probabilities in S2 and S3 Tables.

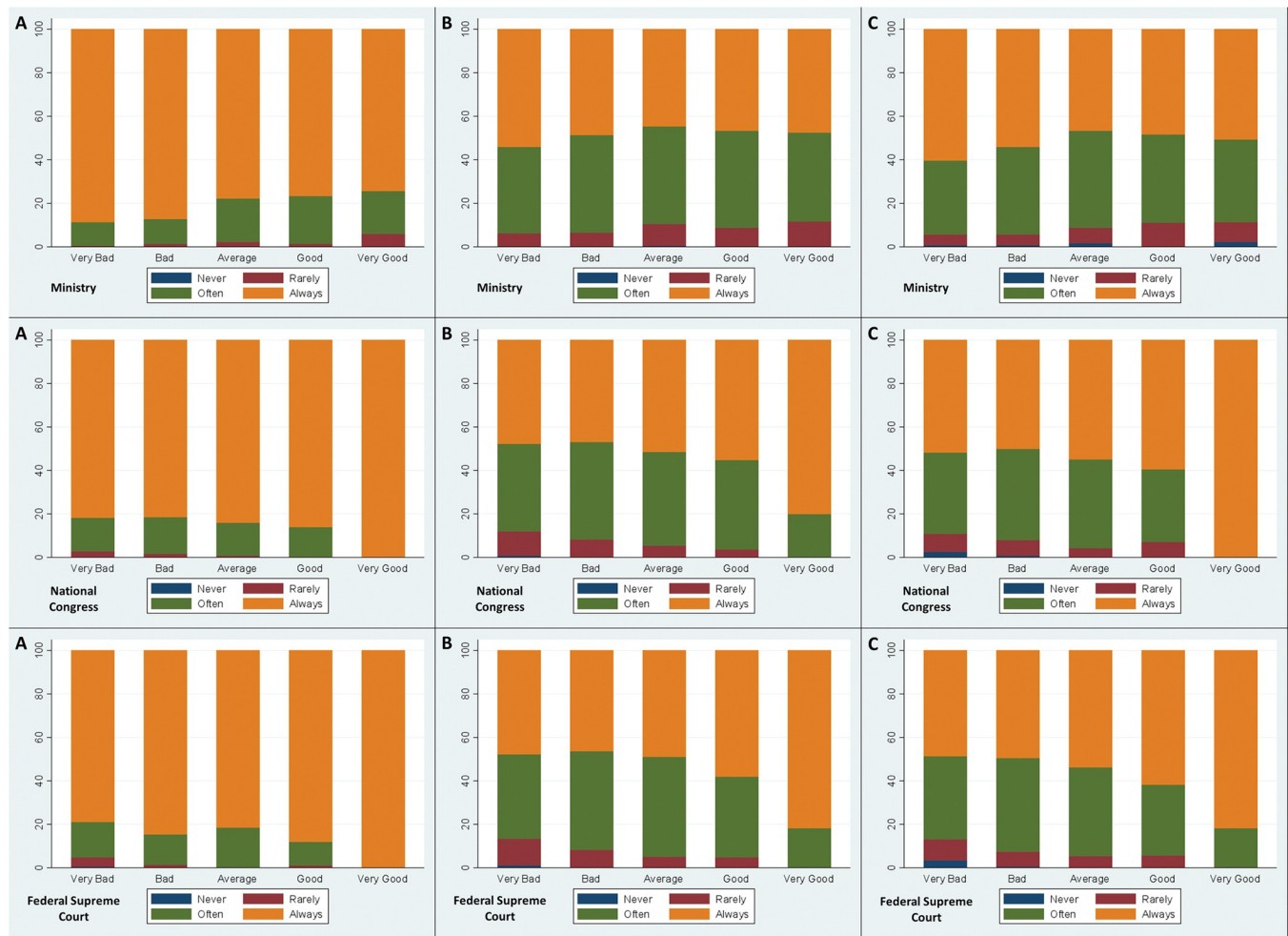

**Fig 2.** Ministry of Health (Executive), National Congress (Legislative) and Federal Supreme Court (Judiciary) x Face Mask (A), Social Distancing (B) and Stay Home (C).

Table 2 summarizes the results of the estimations, considering the association between the performance of the Federal, State and Municipal government and the reported frequency of compliance to the NPIs. We found that an evaluation of the performance of the Federal Government as "Bad" or "Very Bad" is positively associated with compliance to wearing masks and staying at home, with a larger and statistically significant effect for the former. Perceiving that the Federal Government is performing badly is associated with an odds ratio of 1.785 (p<0.01) for wearing masks, 0.961 for social distancing (not significant) and 1.282 (p<0.1) for staying at home. Evaluating the same entity as good or very good, not only has the opposite sign, but it also has a more significant negative impact on the OR of (not) complying with the NPIs. In turn, a "Bad" or "Very Bad" evaluation of the State Government is related to an OR of 0.690 (p<0.05) for wearing masks, 0.801 (p<0.1) for practicing social distancing and 0.665 (p<0.01) for staying at home.

Regarding the evaluations at the municipal level, these were not related to the reported compliance with NPIs at a statistically significant level.

Table 3 summarized the results for the Logit regressions measuring the association between compliance with NPIs and the participants' perception of several national and international institutions as well as with their position on the political spectrum. Regarding the Unified

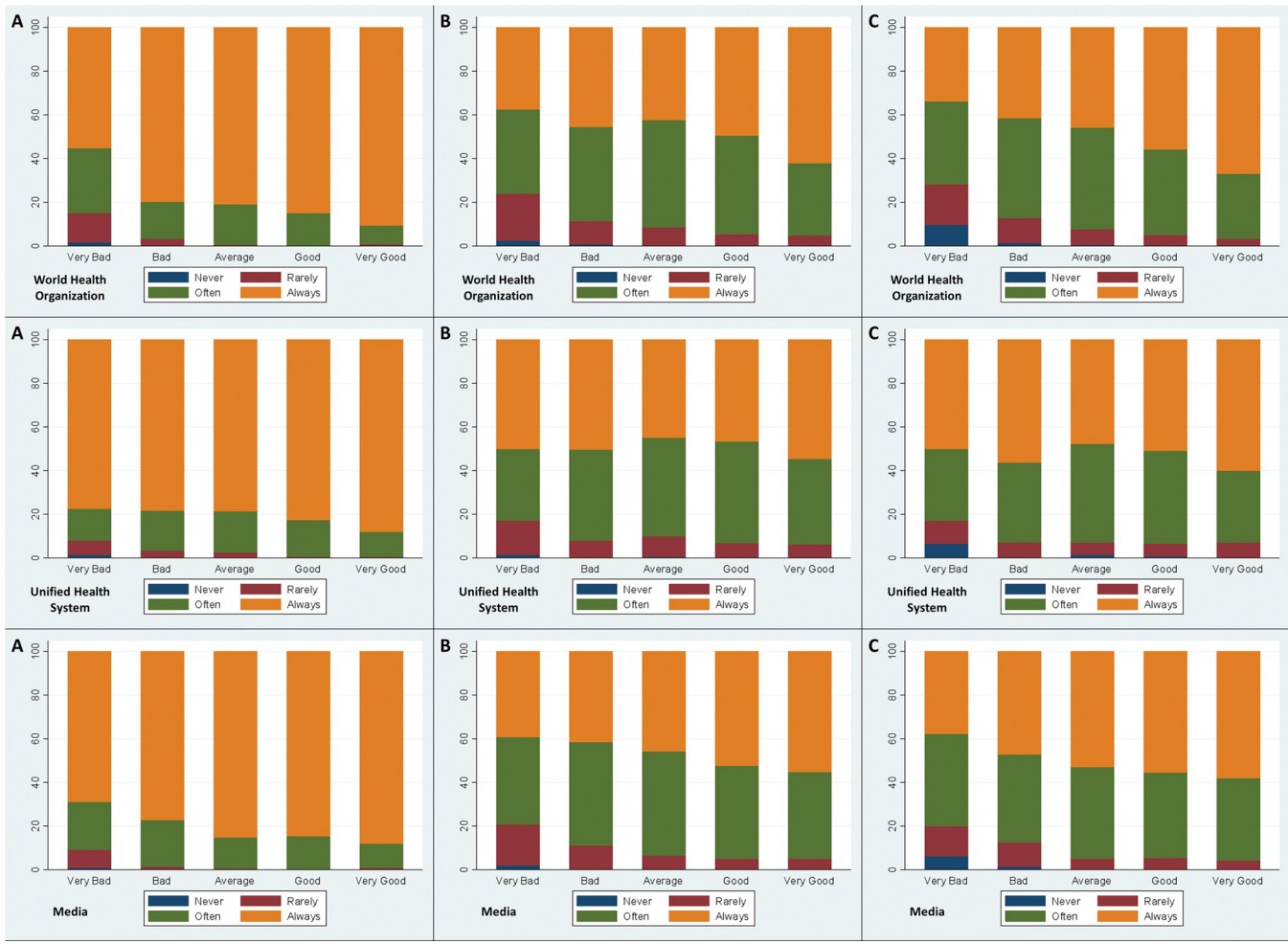

**Fig 3.** World Health Organization (WHO), Unified Health System (SUS) and Media x Face Mask (A), Social Distancing (B) and Stay Home (C).

Health System (SUS), a positive evaluation of the system's performance is associated with more compliance with all NPIs, while having a negative opinion about the SUS is not statistically significant in predicting the probability of any NPI compliance. As for the perception towards the Ministry of Health, the results follow the same trend as before for the Federal Government, a negative evaluation is associated with an OR of 1.869 (p<0.01) on frequency of mask use, 1.187 (p>0.1) for social distancing (not significant) and 1.382 (p<0.01) for staying at home. When it comes to the Brazilian Federal Supreme Court (STF), National Congress and WHO, the association is the opposite, giving a positive evaluation to these institutions increases the odds of complying with the NPIs examined. Finally, responses regarding the political positioning, those who identify more with the far-right political views have lower odds of reporting frequent compliance with NPIs, while those that position themselves more on the left side of the spectrum have more odds of compliance–compliance with using a mask, OR of 2.951 (p<0.01), and staying at home, 1.526 (p<0.05).

## Discussion

This study identified the association between the level of compliance with non-pharmaceutical interventions (face mask use, social distancing and stay-at-home recommendations) and the

**Table 2. Summary table for Ordered Logit regressions: Federal, State and Municipal government spheres x—Face Mask (A), Social Distancing (B) and Stay Home (C)–Odds Ratio.**

|  |  | Mask | Social Dist. | Stay Home |
|---|---|---|---|---|
| **Federal Government** (base group: Fair) | Very bad/bad | 1.785*** | 0.961 | 1.282* |
|  |  | (0.310) | (0.125) | (0.169) |
|  | Good/Very good | 0.433*** | 0.494*** | 0.652** |
|  |  | (0.0935) | (0.0925) | (0.127) |
| **State Government** (base group: Fair) | Very bad/bad | 0.690** | 0.801* | 0.665*** |
|  |  | (0.119) | (0.0999) | (0.0845) |
|  | Good/Very good | 1.095 | 1.050 | 0.996 |
|  |  | (0.228) | (0.150) | (0.143) |
| **Municipal Government** (base group: Fair) | Very bad/bad | 0.930 | 0.972 | 1.202 |
|  |  | (0.160) | (0.120) | (0.153) |
|  | Good/Very good | 1.277 | 1.169 | 1.175 |
|  |  | (0.267) | (0.168) | (0.168) |

Standard errors in parentheses

*** p<0.01, ** p<0.05, * p<0.1

perceived performance of the main governmental and non-governmental institutions, during the COVID-19 outbreak in Brazil. The governmental institutions included in the analysis represent different regional levels (federal, state and municipal), tripartite power (executive—ministry of health; legislative–national congress; and judiciary–federal supreme court), and Unified Health System, and non-governmental institutions, such as WHO and the Media.

Compliance with NPIs and individual's perceptions regarding institutions' performance share some common features. At state levels, there is a negative association between perception of the federal government's performance and NPIs compliance. This result may reflect the personal position of the Brazilian president during the period regarding the COVID-19 pandemic [6, 39]. During the COVID-19 pandemic, the Brazilian former president did not support the application of NPIs and even encouraged the populace not to use them [32, 41]. This is similar to the situation in the United States during President Donald Trump's administration, demonstrating the negative impact of such leadership at the national level [50, 51].

Moreover, the study shows a positive association between the perception of state governments' performance and compliance with NPIs [42], contrary to the results at the federal level. This could be explained by the fact that state governors managed NPIs without federal support [24, 42].

Regarding institutions representing the tripartite power, the results for the Ministry of Health's performance follow the same trend as the one for the federal government, with a negative association between NPIs compliance and perceived performance. This may have been influenced by the instability in the Health Ministry–the minister was replaced several times during the pandemic—as well as with the strong alignment with the President's policy of denial [19]. Indeed, the Ministry of Health was often perceived as being closely aligned with the government. In contrast, the evaluations of the Brazilian federal supreme court and national congress have a positive association with NPIs compliance indicators [52]. As mentioned above, both institutions had their powers reinforced to implement policies, which helped to improve their perceived performance among the population [32, 45].

Concerning WHO, Media and SUS, there is a positive association between performance evaluations and compliance with NPIs [3, 24]. In a world with increasing fake news and misinformation, a partnership between health care international organization and media organizations can be fundamental to improve compliance with NPIs.

**Table 3. Summary table for Ordered Logit regressions: SUS, Ministry of Health, STF, Congress, WHO, media performance and political spectrum x Face Mask (A), Social Distancing (B) and Stay Home (C)–Odds Ratio.**

| | | Mask | Social Dist. | Stay Home |
|---|---|---|---|---|
| **Unified Health System (SUS)** (base group: Fair) | Very bad/bad | 0.887 | 1.044 | 1.116 |
| | | (0.205) | (0.199) | (0.216) |
| | Good/Very good | 1.463** | 1.265** | 1.215* |
| | | (0.231) | (0.150) | (0.140) |
| **Ministry of Health** (base group: Fair) | Very bad/bad | 1.869*** | 1.187 | 1.382*** |
| | | (0.304) | (0.141) | (0.164) |
| | Good/Very good | 0.782 | 1.033 | 1.006 |
| | | (0.133) | (0.146) | (0.143) |
| **Brazilian Federal Supreme Court (STF)** (base group: Fair) | Very bad/bad | 1.065 | 0.806* | 0.737*** |
| | | (0.161) | (0.0899) | (0.0826) |
| | Good/Very good | 1.706** | 1.447** | 1.398* |
| | | (0.433) | (0.245) | (0.240) |
| **National Congress** (base group: Fair) | Very bad/bad | 0.851 | 0.747*** | 0.729*** |
| | | (0.131) | (0.0831) | (0.0811) |
| | Good/Very good | 1.103 | 1.128 | 1.202 |
| | | (0.327) | (0.227) | (0.268) |
| **WHO** (base group: Fair) | Very bad/bad | 0.445*** | 0.704** | 0.523*** |
| | | (0.0882) | (0.120) | (0.0892) |
| | Good/Very good | 1.360** | 1.560*** | 1.609*** |
| | | (0.212) | (0.175) | (0.182) |
| **Media—Performance** (base group: Fair) | Very bad/bad | 0.420*** | 0.560*** | 0.500*** |
| | | (0.0776) | (0.0801) | (0.0727) |
| | Good/Very good | 0.918 | 1.240* | 1.098 |
| | | (0.160) | (0.144) | (0.127) |
| **Political spectrum—Performance** (base group: center) | Far left | 1.550 | 1.643 | 1.767* |
| | | (0.802) | (0.682) | (0.585) |
| | 2 | 2.951*** | 1.262 | 1.526** |
| | | (0.851) | (0.218) | (0.253) |
| | 3 | 1.641** | 1.261 | 1.238 |
| | | (0.361) | (0.190) | (0.185) |
| | 5 | 1.088 | 0.855 | 0.738 |
| | | (0.252) | (0.158) | (0.141) |
| | 6 | 0.514*** | 0.696* | 0.553*** |
| | | (0.124) | (0.143) | (0.115) |
| | Far right | 0.228*** | 0.397*** | 0.389** |
| | | (0.0837) | (0.118) | (0.144) |

Standard errors in parentheses

*** p<0.01, ** p<0.05, * p<0.1

## Conclusions

The findings presented in this paper have four main policy implications: a) the importance of the subnational and local governments in an outbreak context mainly in a decentralized health care system; b) the tripartite power discussion and their limits during exceptional times; c) the importance of international institutions, such as WHO; and d) the reinforcement of the SUS as a key actor not just in the health care delivery but also in the dissemination of public health-related information.

First, during the COVID-19 pandemic, the Brazilian states (subnational) and municipalities (local) were the main authorities responding to the outbreak of disease, with the absence of the federal government, which has a key role in financing the health system [27, 42–44]. This finding contributes directly to other countries with decentralized health systems, such as United States and Mexico [29, 30].

Second, in exceptional times, the legislative and judiciary powers can play a complementary role, to fill in the missing executive action [19, 39, 52]. Third, a partnership and coordination between the media and the World Health Organization can be essential to compete with denial on the parts of both individuals and policy makers [28, 41]. Finally, National Health Systems have an important role, not only in the provision of health care, but also in the communication of public health initiatives [24, 32].

Therefore, the models presented in this paper are valuable case-scenarios in the decentralized health system discussion about institutional performance and public health measures, and the results help to build theory and inform government practices in the public health debate [29–31, 42]. Other health systems in the world, with federative or subnational structure, can also use these findings to improve their performance for NPI compliance. Despite the limitations associated with the observational nature of the paper, we provide evidence of a clear link between NPI and the perception of institutional performance.

Even though causality cannot be claimed, a channel through which perception of institutional performance may influence NPI adherence, is related with the political positioning, and is clearly visible in our findings. This underlying characteristic, specific to each individual, may greatly influence the perception of institutional performance. At the same time, these perceptions will be related with trust in those institutions, which in turn influences adherence to NPIs. Overall, latent characteristics are expected to play a significant role in mediating this relationship.

## Limitations

The study has four major limitations. First, the sample obtained from the survey was not representative of the overall Brazilian population. Despite this limitation, which is common to most online survey settings, our results can still inform policy makers on compliance levels in particular subgroups of the population. Second, results are context specific as they were obtained the situation of a single country. The political landscape has specific characteristics which may not apply elsewhere. Nonetheless, in contexts of high political polarization—as observed recently in the US, for instance—our findings can inform policy makers. Third, the study design implied the selection of a set of institutions. However, there are also other institutions likely to affect the compliance of individuals with NPIs such as religious and social organizations. The forth drawback is the lack of availability of municipal level covariates and the way that aggregating to the municipal level obscures unobserved and unobservable sub-municipal variation. Results are based on the data collected and all of the analysis hinges on correlations. Given the observational nature of this study, even after controlling for multiple factors, causality cannot be claimed.

## Supporting information

**S1 Table. Survey questions.**
(DOCX)

**S2 Table.** Summary table for Ordered Logit regressions: Federal, State and Municipal government spheres x Face Mask (A), Social Distancing (B) and Stay Home (C)–Conditional

Probabilities.
(XLSX)

**S3 Table.** Summary table for Ordered Logit regressions: SUS, Ministry of Health, STF, Congress, WHO, media performance and political spectrum x Face Mask (A), Social Distancing (B) and Stay Home (C)–Conditional Probabilities.
(XLSX)

## Acknowledgments

We would like to thank the support of Brazilian associations and institutions: Brazilian Scientific Editors Association (ABEC), Brazilian Political Science Association (ABCP), Brazilian Association of Collective Health (ABRASCO), among others. Finally, we thank all the people who shared and supported the survey´s dissemination. This research did not receive support from funding agencies in the public, commercial, or not-for-profit sectors.

## Author Contributions

**Conceptualization:** Marco Antonio Catussi Paschoalotto, Eduardo Polena Pacheco Araújo Costa, Sara Valente de Almeida, Joana Gomes da Costa, João Vasco Santos, Claudia Souza Passador, João Luiz Passador, Pedro Pita Barros.

**Data curation:** Marco Antonio Catussi Paschoalotto.

**Formal analysis:** Marco Antonio Catussi Paschoalotto, Joana Cima.

**Investigation:** Marco Antonio Catussi Paschoalotto, Eduardo Polena Pacheco Araújo Costa, Sara Valente de Almeida, Joana Cima.

**Methodology:** Marco Antonio Catussi Paschoalotto, Sara Valente de Almeida, Joana Cima, Joana Gomes da Costa.

**Project administration:** Marco Antonio Catussi Paschoalotto.

**Resources:** Marco Antonio Catussi Paschoalotto.

**Supervision:** Marco Antonio Catussi Paschoalotto, Claudia Souza Passador, João Luiz Passador, Pedro Pita Barros.

**Validation:** Marco Antonio Catussi Paschoalotto, Eduardo Polena Pacheco Araújo Costa, Sara Valente de Almeida, João Vasco Santos.

**Visualization:** Marco Antonio Catussi Paschoalotto, Eduardo Polena Pacheco Araújo Costa, Sara Valente de Almeida.

**Writing – original draft:** Marco Antonio Catussi Paschoalotto, Eduardo Polena Pacheco Araújo Costa, Sara Valente de Almeida, Joana Cima, Joana Gomes da Costa, João Vasco Santos, Claudia Souza Passador, João Luiz Passador, Pedro Pita Barros.

**Writing – review & editing:** Marco Antonio Catussi Paschoalotto, Eduardo Polena Pacheco Araújo Costa, Sara Valente de Almeida, Joana Cima, Joana Gomes da Costa, João Vasco Santos, Claudia Souza Passador, João Luiz Passador, Pedro Pita Barros.

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
