## [Decision Letter · Decision Letter 0]

12 Dec 2022

PONE-D-22-28667The perception of institutional performance and the adherence to non-pharmaceutical interventions: how does it affect public health in a fragmented health system?PLOS ONE

Dear Dr. Paschoalotto,

Thank you for submitting your manuscript to PLOS ONE. After careful consideration, we feel that it has merit but does not fully meet PLOS ONE’s publication criteria as it currently stands. Therefore, we invite you to submit a revised version of the manuscript that addresses the points raised during the review process.

We look forward to receiving your revised manuscript.

Kind regards,

Ivan Filipe de Almeida Lopes Fernandes, Ph.D.

Academic Editor

PLOS ONE

Journal Requirements:

We will update your Data Availability statement to reflect the information you provide in your cover letter.\\

Additional Editor Comments:

The article's theme is relevant, but there are already published articles that investigate the subnational politics of COVID-19 in Brazil, which should be further reviewed.

Besides, the paper should be more engaged in explaining the variation in institutional trust and NPI compliances. A more considered assessment of the limits of the study should be made, including the extent to which the sample allows generalizations to be made about the Brazilian population and the scope of these generalizations.

Lastly, the study should discuss the frictions between the credibility of institutions and the political dynamics that occurred in COVID-19 generally and, in particular, the Brazilian dynamics that produced a clash among subnational authorities and the federal government.

Reviewers' comments:

Reviewer's Responses to Questions

**Comments to the Author**

1. Is the manuscript technically sound, and do the data support the conclusions?

Reviewer #1: Yes

Reviewer #2: Partly

2. Has the statistical analysis been performed appropriately and rigorously? 

Reviewer #1: Yes

Reviewer #2: Yes

3. Have the authors made all data underlying the findings in their manuscript fully available?

Reviewer #1: Yes

Reviewer #2: No

4. Is the manuscript presented in an intelligible fashion and written in standard English?

Reviewer #1: Yes

Reviewer #2: No

5. Review Comments to the Author

Reviewer #1: I have several recommendations that I think will improve the overall contribution, which I already find compelling, but suggest some revisions before publication. First, I suggest adding a “limitations” section along the lines of what we’ve seen recently in many public health and medical journal publications surrounding covid-19. Most of these limitations are methodological, which I think is completely acceptable given the real-time data collection and analysis in question.

causal identification is very difficult in this context, which I think is reasonable and currently common in the top medical journals for work on covid. However, I suggest moderating some of the claims in the paper and describing why the models presented represent the best-case scenario in the current climate and how the results can still help build theory and inform government practice. Another limitation is simply the lack of municipal level covariates available and/or the way that aggregating to the municipal level obscures unobserved and (probably) unobservable sub-municipal variation.

Next, I suggest defending the case selection a little more thoroughly. For example, are the results from Brazil generalizable beyond the country? Why/why not? I agree with the authors’ rationale and I think they should make an even larger claim: that brazil, due to data availability and variation at all levels of government for perceptions of performance and NPI, is the only country where they could plausibly test hypotheses against such broad, deep data.

Defending the timeframe under consideration should also be part of the next revision. I see lots of reasons to focus on the timeframe under consideration in the article, but I would like to see those reasons articulated thoroughly. For example, because the covid response shifted from NPIs to vaccines after 2020-2021.

Next, I think the paper will have a greater impact if placed more thoroughly in the global and regional comparative context. For example, there is more recent literature on Brazil and Mexico and on subnational covid issues in Latin America in general that should be included in the review and with which this article can enter into conversation. These are just a few examples below that I think should be included, including one from PLOS-One, but there are many others that would help this paper engage a burgeoning global literature in the area of the study.

Testa, Paul F., Richard Snyder, Eva Rios, Eduardo Moncada, Agustina Giraudy, and Cyril Bennouna. "Who Stays at Home? The Politics of Social Distancing in Brazil, Mexico, and the United States during the COVID-19 Pandemic." Journal of Health Politics, Policy and Law (2021).

Knaul, F. M., Touchton, M., Arreola-Ornelas, H., Atun, R., Anyosa, R. J. C., Frenk, J., ... & Victora, C. (2021). Punt politics as failure of health system stewardship: evidence from the COVID-19 pandemic response in Brazil and Mexico. The Lancet Regional Health-Americas, 4, 100086.

Castro, Marcia C., Sun Kim, Lorena Barberia, Ana Freitas Ribeiro, Susie Gurzenda, Karina Braga Ribeiro, Erin Abbott, Jeffrey Blossom, Beatriz Rache, and Burton H. Singer. "Spatiotemporal pattern of COVID-19 spread in Brazil." Science 372, no. 6544 (2021): 821-826.

Knaul, Felicia, Héctor Arreola-Ornelas, Thalia Porteny, Michael Touchton, Mariano Sánchez-Talanquer, Óscar Méndez, Salomón Chertorivski et al. "Not far enough: Public health policies to combat COVID-19 in Mexico’s states." Plos one 16, no. 6 (2021): e0251722.

The rest of the paper is very well-done. By engaging with the recent literature on the specific subject, the paper will be easier to find for scholars and policymakers working in the area.

Finally, I suggest going through the paper carefully in English, or with the help of a native speaker. The writing is well-done, but there are minor errors or awkward phrasing throughout.

For example, the title would be better as “Perceptions of institutional performance and adherence to non-pharmaceutical interventions: how performance perceptions and policy adherence affect public health in a fragmented health system”

The sub-title would be better as “The relationship between institutional performance and public health interventions”

Reviewer #2: This manuscript constitutes a rigorous attempt to examine the relationship between trust in government institutions and compliance with public health guidelines. It has the potential to offer an important contribution to our understanding of the Brazilian case, in the context of the Covid-19 pandemic. I will enumerate my suggestions and concerns, which amount to some measure of revisions -- between minor and major. There are some methodological aspects and other points of clarification.

1. Given that Plos One does not copy edit approved manuscripts, it is important to submit this manuscript to a professional copy editor. There are issues of sintaxe that compromise the comprehension of the text -- already considering a specialized audience.

2. I would like to have seen more details on the online survey. The questionnaire itself, the presenting/introducing email, where the sending email came from, the accompanying instructions. These conditions may predispose the respondent to: a) answer, not answer; b) answer in a certain way.

3. Another point regarding methodology, as it relates to the online survey: is it possible to draw a representative sample of the responses that were collected? If so, how does this representative sample compares to the actual sample that was used for the statistical analysis? The authors discuss the overrepresentation of the State of Sao Paulo in the responses. What does that mean? How does this characteristic of the sample affect the results? Are the majority of the authors and their institutions based in Sao Paulo?

4. I believe it is important to clarify the relationship between the SUS and the branches of government in Brazil. I am not convinced that respondents are able -- on average -- to disentangle the intricacies behind the governance structure of this system. This may compromise their responses in this case. In my view, the most important, the central findings of the analysis rest with the federal v. state governments cleavage -- with respect to public health policies in the context of Covid-19. A little more on the alliance amongst state governors here, together with background on the protagonism of the Brazilian Supreme Court would help paint a more complete picture of the Brazilian case.

5. Still on clarification, the authors should state what they understand by "non-pharmaceutical interventions" upfront as well as what they label "compliance." These definitions are important as they will guide the readers as they navigate the following sections of the manuscript. They are also important because they represent challenges for monitoring and enforcement, thus the role of trust in government as a main engine behind the individual decision to comply. A more detailed discussion could even touch on the legitimacy of the NPI measure at hand, given the source of authority behind this measure.

6. Given the complexity of interpreting odds ration results, the authors should present a table of conditional probabilities for some (important) moments of the data. This is the case for the main findings and could be desegregated by individual NPI measure.

7. Finally, I would like to see a more explicit discussion of the importance of the "latent" characteristic/choice. What does the manuscript gain with this approach?

6. PLOS authors have the option to publish the peer review history of their article (what does this mean?). If published, this will include your full peer review and any attached files.

Reviewer #1: No

Reviewer #2: No

---

## [Author Response · Author response to Decision Letter 0]

25 Jan 2023

PONE-D-22-28667

Perceptions of institutional performance and compliance to non-pharmaceutical interventions: how performance perceptions and policy compliance affect public health in a fragmented health system

Additional requirements

Revision: Thank you very much. We followed the PLOS ONE's style requirements and changed the paper accordingly to address them. You can now see the paper complying with the PLOS ONE's style requirements.

Revision: Thank you for your comment. We inserted in the manuscript (materials and methods section) a new text informing the ethical approvement and proceedings, and the participant consent information.

“The survey was approved on 23 November 2020, by the Research Ethics Committee at NOVA School of Business and Economics (Portugal) through a Scientific Council’s president letter. In the online survey’s first page, the participants read and accepted the research characteristics and information, assurances of anonymity assurance, data protection, and a consent form.”

Revision: Thank you very much for reminding us of these points. We had also not included information about the contributorship, funding and competing interest. To solve all these issues, we inserted in the first page the following text:

“Contributorship: The authors contributed according to the following roles: i) Conception and design of the work: All authors participated (task led by Marco Paschoalotto); ii) Data collection: All authors participated (task led by Marco Paschoalotto); iii) Data analysis and interpretation: Task performed by Marco Paschoalotto, Eduardo Costa, Sara Valente, Joana Cima and Joana Gomes da Costa; iv) Drafting the article: All authors participated (task led by Marco Paschoalotto with major contributions from Eduardo Costa, Sara Valente, Joana Cima, João Vasco Santos and Joana Gomes da Costa); v) Critical revision of the article: All authors participated (task led by Marco Paschoalotto); vi) Final approval of the version to be submitted: All the authors approved the submitted version.

Data Availability: The dataset is anonymous and does not allow the identification of individual respondents. However, based on the informed consent signed by respondents and on the authorization granted by NOVA School of Business and Economics Ethics Committee, data cannot be shared in a public repository to respect data protection regulation. A minimal data set for replication of the study findings is available upon request to the NOVA School of Business and Economics Ethics Committee (research.office@novasbe.pt) and the corresponding author.

Funding: there are no financial relationships relevant to this article to disclose.

Competing interests: The authors have declared that no competing interests exist.”

Revision: Thank you for your comment. We inserted a new text informing the ethical approvement and proceedings, and the participant consent information.

“The survey was approved on 23 November 2020, by the Research Ethics Committee at NOVA School of Business and Economics (Portugal) through a Scientific Council’s president letter. In the online survey’s first page, the participants read and accepted the research characteristics and information, assurances of anonymity assurance, data protection, and a consent form.”

Editor Comments

The article's theme is relevant, but there are already published articles that investigate the subnational politics of COVID-19 in Brazil, which should be further reviewed.

Besides, the paper should be more engaged in explaining the variation in institutional trust and NPI compliances. A more considered assessment of the limits of the study should be made, including the extent to which the sample allows generalizations to be made about the Brazilian population and the scope of these generalizations.

Lastly, the study should discuss the frictions between the credibility of institutions and the political dynamics that occurred in COVID-19 generally and, in particular, the Brazilian dynamics that produced a clash among subnational authorities and the federal government.

Revision: Thank you for your comment and pointing out these relevant aspects. In response to this and the reviewers’ comments below, we’ve added several new references on the politics of COVID-19 in Brazil that we describe in the new introduction section. These new references are highlighted in yellow in the the new version of the manuscript reference list. We’ve also developed our limitations section further to include the fact that our results are context specific and not easily generalizable, and the lack of availability of municipal level covariates that limits our analysis. Regarding the political dynamics that occurred during the pandemic, we also added a paragraph in the introduction describing changes in autonomy and political powers that resulted from frictions between different institutions, to give a general overview of how the political situation changed during the course of this challenging period.

Reviewer 1

I have several recommendations that I think will improve the overall contribution, which I already find compelling, but suggest some revisions before publication. First, I suggest adding a “limitations” section along the lines of what we’ve seen recently in many public health and medical journal publications surrounding covid-19. Most of these limitations are methodological, which I think is completely acceptable given the real-time data collection and analysis in question.

Revision: Dear reviewer 1, thank you for the comment. We agree with your point, and thus we created a new section called “limitations”. Based on your next comment and the editor’s comment, we also included other relevant limitations. You can see the modifications in the manuscript described below:

“Limitations 

The study has four major limitations. First, the sample obtained from the survey was not representative of the overall Brazilian population. Despite this limitation, which is common to most online survey settings, our results can still inform policy makers on compliance levels in particular subgroups of the population. Second, results are context specific as they were obtained the situation of a single country. The political landscape has specific characteristics which may not apply elsewhere. Nonetheless, in contexts of high political polarization - as observed recently in the US, for instance - our findings can inform policy makers. Third, the study design implied the selection of a set of institutions. However, there are also other institutions likely to affect the compliance of individuals with NPIs such as religious and social organizations. The forth drawback is the lack of availability of municipal level covariates and the way that aggregating to the municipal level obscures unobserved and unobservable sub-municipal variation. Results are based on the data collected and all of the analysis hinges on correlations. Given the observational nature of this study, even after controlling for multiple factors, causality cannot be claimed.”

Causal identification is very difficult in this context, which I think is reasonable and currently common in the top medical journals for work on covid. However, I suggest moderating some of the claims in the paper and describing why the models presented represent the best-case scenario in the current climate and how the results can still help build theory and inform government practice. Another limitation is simply the lack of municipal level covariates available and/or the way that aggregating to the municipal level obscures unobserved and (probably) unobservable sub-municipal variation.

Revision: Dear reviewer 2, thank you for the 2 points described here. First, we followed your advice and described why our results were important and the best scenario. You can find the new paragraph in the Conclusion section of the new manuscript:

"Therefore, the models presented in this paper are valuable case-scenarios in the fragmented health system discussion about institutional performance and public health measures, and the results help to build theory and inform government practices in the public health debate (30–32,43). Other health systems in the world, with federative or subnational structure, can also use these findings to improve their performance for NPI compliance. "

Also, we included the limitation proposed by you:

“The forth drawback is the lack of availability of municipal level covariates and the way that aggregating to the municipal level obscures unobserved and unobservable sub-municipal variation. Results are based on the data collected and all of the analysis hinges on correlations. Given the observational nature of this study, even after controlling for multiple factors, causality cannot be claimed.”

Next, I suggest defending the case selection a little more thoroughly. For example, are the results from Brazil generalizable beyond the country? Why/why not? I agree with the authors’ rationale and I think they should make an even larger claim: that brazil, due to data availability and variation at all levels of government for perceptions of performance and NPI, is the only country where they could plausibly test hypotheses against such broad, deep data.

Revision: Dear reviewer 1, thank you for calling us up to defend our case selection. Based on your comments, we added the following sentence in the limitations of the new manuscript:

"Second, results are context specific as they were obtained the situation of a single country. The political landscape has specific characteristics which may not apply elsewhere. Nonetheless, in contexts of high political polarization - as observed recently in the US, for instance - our findings can inform policy makers”

Defending the timeframe under consideration should also be part of the next revision. I see lots of reasons to focus on the timeframe under consideration in the article, but I would like to see those reasons articulated thoroughly. For example, because the covid response shifted from NPIs to vaccines after 2020-2021.

Revision: Dear reviewer 1, thank you for advising us to defend our timeframe. Following your comment, we changed the text in the Data Collection section and included the following text:

“The time interval regarding data collection was important because: i) it was applied before the second largest (and deadliest) COVID-19 wave, ii) the survey was responded immediately before the introduction of the first COVID-19 vaccine in Brazil, CoronaVac, in March 2021, iii) it marked a change in the strategy addressing COVID-19, shifting from NPIs to vaccination, and iv) it marked the consolidation of the population’ perception towards public health measures, which is seen after 9 months of introduction.”

Next, I think the paper will have a greater impact if placed more thoroughly in the global and regional comparative context. For example, there is more recent literature on Brazil and Mexico and on subnational covid issues in Latin America in general that should be included in the review and with which this article can enter into conversation. These are just a few examples below that I think should be included, including one from PLOS-One, but there are many others that would help this paper engage a burgeoning global literature in the area of the study.

Testa, Paul F., Richard Snyder, Eva Rios, Eduardo Moncada, Agustina Giraudy, and Cyril Bennouna. "Who Stays at Home? The Politics of Social Distancing in Brazil, Mexico, and the United States during the COVID-19 Pandemic." Journal of Health Politics, Policy and Law (2021).

Knaul, F. M., Touchton, M., Arreola-Ornelas, H., Atun, R., Anyosa, R. J. C., Frenk, J., ... & Victora, C. (2021). Punt politics as failure of health system stewardship: evidence from the COVID-19 pandemic response in Brazil and Mexico. The Lancet Regional Health-Americas, 4, 100086.

Castro, Marcia C., Sun Kim, Lorena Barberia, Ana Freitas Ribeiro, Susie Gurzenda, Karina Braga Ribeiro, Erin Abbott, Jeffrey Blossom, Beatriz Rache, and Burton H. Singer. "Spatiotemporal pattern of COVID-19 spread in Brazil." Science 372, no. 6544 (2021): 821-826.

Knaul, Felicia, Héctor Arreola-Ornelas, Thalia Porteny, Michael Touchton, Mariano Sánchez-Talanquer, Óscar Méndez, Salomón Chertorivski et al. "Not far enough: Public health policies to combat COVID-19 in Mexico’s states." Plos one 16, no. 6 (2021): e0251722.

Revision: Dear reviewer 1, thank you for the comment to debate more thoroughly in the global and regional comparative context. Following your comment, we added the following texts and new references (highlighted in yellow):

“Regarding the American continent for example, it was described that the USA, Brazil, and Mexico had different performances in restricting their populations’ mobility through different public health approaches, but all experienced problems of coordination (30). These problems were mainly seen in Brazil and Mexico (31). Mexico did not present an uniform national response, with the states varying in their implementation of NPIs, compromising the national health system governance (32). Brazil also did not present a national coordinated response in the public health measures to combat the COVID-19 pandemic, with the virus spreading differently across the states (33,34).”

“Therefore, the models presented in this paper are valuable case-scenarios in the fragmented health system discussion about institutional performance and public health measures, and the results help to build theory and inform government practices in the public health debate (30–32,43). Other health systems in the world, with federative or subnational structure, can also use these findings to improve their performance for NPI compliance.”

Finally, I suggest going through the paper carefully in English, or with the help of a native speaker. The writing is well-done, but there are minor errors or awkward phrasing throughout.

For example, the title would be better as “Perceptions of institutional performance and adherence to non-pharmaceutical interventions: how performance perceptions and policy adherence affect public health in a fragmented health system”.

The sub-title would be better as “The relationship between institutional performance and public health interventions”.

Revision: Dear reviewer 1, thank you for your suggestion to change the title and subtitle. We followed your advice and changed the title and subtitle accordingly (see title page). Also, in this revision, we used a professional “proofreading” to revise the English writing in our paper. You can see the modifications over all the text. 

Reviewer 2

1. Given that Plos One does not copy edit approved manuscripts, it is important to submit this manuscript to a professional copy editor. There are issues of sintaxe that compromise the comprehension of the text -- already considering a specialized audience.

Revision: Dear reviewer 2, following your advice, we used a professional “proofreading” to revise the English writing in our paper. You can see the modifications over all the text. 

2. I would like to have seen more details on the online survey. The questionnaire itself, the presenting/introducing email, where the sending email came from, the accompanying instructions. These conditions may predispose the respondent to: a) answer, not answer; b) answer in a certain way.

Revision: Dear reviewer 2, thank you for asking more information. Following your advice, we added as supporting information (S1 Table) the survey questions with the percentage of answers to each option. 

Also, in the “materials and methods” section, we added the following text regarding the questionnaire, consent, and information.

“The survey was approved on 23 November 2020, by the Research Ethics Committee at NOVA School of Business and Economics (Portugal) through a Scientific Council’s president letter. In the online survey’s first page, the participants read and accepted the research characteristics and information, assurances of anonymity assurance, data protection, and a consent form.”

3. Another point regarding methodology, as it relates to the online survey: is it possible to draw a representative sample of the responses that were collected? If so, how does this representative sample compares to the actual sample that was used for the statistical analysis? The authors discuss the overrepresentation of the State of Sao Paulo in the responses. What does that mean? How does this characteristic of the sample affect the results? Are the majority of the authors and their institutions based in Sao Paulo?

Revision: Dear reviewer 2, thank you for the questions. To better answer the questions, we are going to do it one by one:

1) is it possible to draw a representative sample of the responses that were collected?

A: Table 1 presents the sample characteristics, as well as the S1 table the percentage of the variables analyzed. It would not be possible with the answers collected to draw a representative sample, due to the number of answers, absence of answers from all the states, and sociodemographic characteristics. 

2) If so, how does this representative sample compares to the actual sample that was used for the statistical analysis? 

A: Based on the principle that “It would not be possible with the answers collected to draw a representative sample”, we could not compare them. 

3) The authors discuss the overrepresentation of the State of Sao Paulo in the responses. What does that mean? How does this characteristic of the sample affect the results? Are the majority of the authors and their institutions based in Sao Paulo?

A: The overrepresentation of the State of Sao Paulo in the responses means that we have more answers from the São Paulo State (67%) than the other states, considering that the São Paulo State population concentrates approximately 21% of the Brazilian population. This is common to many observational studies and it happens because of our network bias and absence of financial support to hire a representative sample (yes, most part of the authors are from the São Paulo State). 

Finally, in our “data collection” subsection we state that “Despite de fact that this is not a representative sample”, informing that is not a representative sample, which means has its own problems. Also, in the “limitations” subsection, we say: “the sample obtained from the survey was not representative of the overall Brazilian population. Even though this limitation is common to most online survey settings, our results can still inform policy makers on compliance levels in particular subgroups of the population”.

4. I believe it is important to clarify the relationship between the SUS and the branches of government in Brazil. I am not convinced that respondents are able -- on average -- to disentangle the intricacies behind the governance structure of this system. This may compromise their responses in this case. In my view, the most important, the central findings of the analysis rest with the federal v. state governments cleavage -- with respect to public health policies in the context of Covid-19. A little more on the alliance amongst state governors here, together with background on the protagonism of the Brazilian Supreme Court would help paint a more complete picture of the Brazilian case.

Revision: Dear reviewer 2, thank you for the clarification. We agree with you and to better align the state governors discussion, together with background on the protagonism of the Brazilian Supreme Court, we changed the text in the introduction and added the following phrases:

“In response to this attitude, the judiciary and legislative powers, the Federal Supreme Court (STF) and the National Congress, took political measures to supplement the federal government action (43–45). The states and municipalities acquired more autonomy to act against the COVID-19 pandemic, due to a Federal Supreme Court (STF) decision, and the number of social programs to provide financial support to the Brazilian society increased, due to National Congress emergency legislation (28,33,43,46). The states’ governors and respective state health secretaries aligned their responses, supported by the National Council of State Health Secretaries (CONASS), to implement public health measures to combat the COVID-19 spread (34).”

5. Still on clarification, the authors should state what they understand by "non-pharmaceutical interventions" upfront as well as what they label "compliance." These definitions are important as they will guide the readers as they navigate the following sections of the manuscript. They are also important because they represent challenges for monitoring and enforcement, thus the role of trust in government as a main engine behind the individual decision to comply. A more detailed discussion could even touch on the legitimacy of the NPI measure at hand, given the source of authority behind this measure.

Revision: Dear reviewer 2, thank you for the recommendation to define and clarify non-pharmaceutical interventions and compliance definitions used. To address it, we added the following phrase:

“As i) NPIs we consider individual (e.g. face mask), environmental (e.g. ventilation of spaces) and population (e.g. social distancing and stay-at-home/quarantine measures) interventions, and as ii) compliance we consider the self-reported level of compliance (Never, Rarely, Frequently and Always) with NPIs (8).”

6. Given the complexity of interpreting odds ration results, the authors should present a table of conditional probabilities for some (important) moments of the data. This is the case for the main findings and could be desegregated by individual NPI measure.

Revision: Dear reviewer 2, thank you for the advice. We accepted your proposal, and we created 2 tables with the conditional probabilities. The new tables can be found in the supporting information section (S2 Table and S3 Table). We also included the following phrase in the Regressions section:

“We also present the institutions’ performance odds and tables of conditional probabilities in the Supplementary material, S2 Table and S3 Table.”

7. Finally, I would like to see a more explicit discussion of the importance of the "latent" characteristic/choice. What does the manuscript gain with this approach?

Revision: Dear reviewer 2, thank you for the comment. We improved the discussion on the importance of the "latent" characteristic/choice adding the following text in the Conclusions section:

“Even though causality cannot be claimed, a channel through which perception of institutional performance may influence NPI adherence, is related with the political positioning, and is clearly visible in our findings. This underlying characteristic, specific to each individual, may greatly influence the perception of institutional performance. At the same time, these perceptions will be related with trust in those institutions, which in turn influences adherence to NPIs. Overall, latent characteristics are expected to play a significant role in mediating this relationship.”

---

## [Decision Letter · Decision Letter 1]

12 Apr 2023

PONE-D-22-28667R1Perceptions of institutional performance and compliance to non-pharmaceutical interventions: how performance perceptions and policy compliance affect public health in a fragmented health systemPLOS ONE

Dear Dr. Paschoalotto,

Thank you for submitting your manuscript to PLOS ONE. After careful consideration, we feel that it has merit but does not fully meet PLOS ONE’s publication criteria as it currently stands. Therefore, we invite you to submit a revised version of the manuscript that addresses the points raised during the review process.

Thank you very much for your efforts in answering the reviewers' requests and comments. The manuscript has been substantially improved and is almost ready to be published by the journal.

However, I would like to request some important minor corrections to specific issues in the text.

1) SUS is not fragmented but decentralized. So it was clear the effort that was made to adjust the text to this issue; however, it still missed some indications of the SUS as a fragmented system, which is quite a mistaken assessment, since there is a huge effort of federative coordination within the SUS structure, which is the largest existing public health system.

a. Some indications need to be corrected on the title, pages 3 (abstract) and 16 (conclusions)

2) Brazil is not a low-middle income country but an Upper Middle Income Country, as can be seen on the World Bank's website (https://data.worldbank.org/?locations=BR-XT)

3) It is important to make it clear to the reader that the sample is convenient and not representative. Thus, at the end of the introduction, I ask you to insert the word "convenience" in the following sentence.

For this purpose, we rely on an online "convenience" national survey to explore (...)

We look forward to receiving your revised manuscript.

Kind regards,

Ivan Filipe de Almeida Lopes Fernandes, Ph.D.

Academic Editor

PLOS ONE

Journal Requirements:

Additional Editor Comments:

Dear Authors

Thank you very much for your efforts in answering the reviewers' requests and comments. The manuscript has been substantially improved and is almost ready to be published by the journal.

However, I would like to request some important minor corrections to specific issues in the text

1) SUS is not fragmented but decentralized. So it was clear the effort that was made to adjust the text to this issue; however, it still missed some indications of the SUS as a fragmented system, which is quite a mistaken assessment, since there is a huge effort of federative coordination within the SUS structure, which is the largest existing public health system.

a. Some indications need to be corrected on the title, pages 3 (abstract) and 16 (conclusions)

2) Brazil is not a low-middle income country but an Upper Middle Income Country, as can be seen on the World Bank's website (https://data.worldbank.org/?locations=BR-XT)

3) It is important to make it clear to the reader that the sample is convenient and not representative. Thus, at the end of the introduction, I ask you to insert the word "convenience" in the following sentence.

For this purpose, we rely on an online "convenience" national survey to explore (...)

Regards,

Reviewers' comments:

Reviewer's Responses to Questions

**Comments to the Author**

1. If the authors have adequately addressed your comments raised in a previous round of review and you feel that this manuscript is now acceptable for publication, you may indicate that here to bypass the “Comments to the Author” section, enter your conflict of interest statement in the “Confidential to Editor” section, and submit your "Accept" recommendation.

Reviewer #1: All comments have been addressed

Reviewer #2: All comments have been addressed

2. Is the manuscript technically sound, and do the data support the conclusions?

Reviewer #1: Yes

Reviewer #2: Yes

3. Has the statistical analysis been performed appropriately and rigorously? 

Reviewer #1: Yes

Reviewer #2: Yes

4. Have the authors made all data underlying the findings in their manuscript fully available?

Reviewer #1: Yes

Reviewer #2: Yes

5. Is the manuscript presented in an intelligible fashion and written in standard English?

Reviewer #1: Yes

Reviewer #2: Yes

6. Review Comments to the Author

Reviewer #1: the paper now incorporates all of my suggestions- the new material improves the submission and makes a larger contribution than in the first version

Reviewer #2: (No Response)

7. PLOS authors have the option to publish the peer review history of their article (what does this mean?). If published, this will include your full peer review and any attached files.

Reviewer #1: No

Reviewer #2: No

---

## [Author Response · Author response to Decision Letter 1]

17 Apr 2023

PONE-D-22-28667R1

Perceptions of institutional performance and compliance to non-pharmaceutical interventions: how performance perceptions and policy compliance affect public health in a decentralized health system

Editor’s/Reviewers’ requirements

1) SUS is not fragmented but decentralized. So it was clear the effort that was made to adjust the text to this issue; however, it still missed some indications of the SUS as a fragmented system, which is quite a mistaken assessment, since there is a huge effort of federative coordination within the SUS structure, which is the largest existing public health system.

a. Some indications need to be corrected on the title, pages 3 (abstract) and 16 (conclusions)

Revision: Dear Editor and Reviewer, thank you very much for the contribution. Following your advice, we changed all the terms (4) “fragmented” to “decentralized” in the text (title, abstract and 2 in the conclusions). 

2) Brazil is not a low-middle income country but an Upper Middle Income Country, as can be seen on the World Bank's website (https://data.worldbank.org/?locations=BR-XT)

Revision: Dear Editor and Reviewer, thank you very much for the contribution. Following your advice, we changed the term “low middle-income country” to “upper middle-income country” in the text (page 5).

3) It is important to make it clear to the reader that the sample is convenient and not representative. Thus, at the end of the introduction, I ask you to insert the word "convenience" in the following sentence.

For this purpose, we rely on an online "convenience" national survey to explore (...)

Revision: Dear Editor and Reviewer, thank you very much for the contribution. Following your advice, we changed the sentence to “For this purpose, we rely on an online convenient national survey to explore (…)”. 

Journal requirements

Revision: Dear journal, we revised our reference list and citations, and it was correct, so no changes were made.

---

## [Editor Report · Decision Letter 2]

19 Apr 2023

Perceptions of institutional performance and compliance to non-pharmaceutical interventions: how performance perceptions and policy compliance affect public health in a decentralized health system

PONE-D-22-28667R2

Dear Dr. Paschoalotto,

We’re pleased to inform you that your manuscript has been judged scientifically suitable for publication and will be formally accepted for publication once it meets all outstanding technical requirements.

Kind regards,

Ivan Filipe de Almeida Lopes Fernandes, Ph.D.

Academic Editor

PLOS ONE
---

## [Editor Report · Acceptance letter]

5 May 2023

PONE-D-22-28667R2 

Perceptions of institutional performance and compliance to non-pharmaceutical interventions: how performance perceptions and policy compliance affect public health in a decentralized health system 

Dear Dr. Paschoalotto:

I'm pleased to inform you that your manuscript has been deemed suitable for publication in PLOS ONE. Congratulations! Your manuscript is now with our production department. 

Kind regards, 

on behalf of

Dr. Ivan Filipe de Almeida Lopes Fernandes 

Academic Editor

PLOS ONE